# Efficient ANN-SNN Conversion with Error Compensation Learning

Chang Liu [* 1]  Jiangrong Shen [* 2 3]  Xuming Ran [4]  Mingkun Xu [5]  Qi Xu [† 1]  Yi Xu [1]  Gang Pan [3 6]

## Abstract

Artificial neural networks (ANNs) have demonstrated outstanding performance in numerous tasks, but deployment in resource-constrained environments remains a challenge due to their high computational and memory requirements. Spiking neural networks (SNNs) operate through discrete spike events and offer superior energy efficiency, providing a bio-inspired alternative. However, current ANN-to-SNN conversion often results in significant accuracy loss and increased inference time due to conversion errors such as clipping, quantization, and uneven activation. This paper proposes a novel ANN-to-SNN conversion framework based on error compensation learning. We introduce a learnable threshold clipping function, dual-threshold neurons, and an optimized membrane potential initialization strategy to mitigate the conversion error. Together, these techniques address the clipping error through adaptive thresholds, dynamically reduce the quantization error through dual-threshold neurons, and minimize the non-uniformity error by effectively managing the membrane potential. Experimental results on CIFAR-10, CIFAR-100, ImageNet datasets show that our method achieves high-precision and ultra-low latency among existing conversion methods. Using only two time steps, our method significantly reduces the inference time while maintains competitive accuracy of 94.75% on CIFAR-10 dataset under ResNet-18 structure. This research promotes the practical application of SNNs on low-power hardware, making efficient real-time processing possible.

*Equal contribution [1]School of Computer Science and Technology, Dalian University of Technology, Dalian, China [2]Faculty of Electronic and Information Engineering, Xi'an Jiaotong University [3]State Key Lab of Brain-Machine Intelligence, Zhejiang University [4]National University of Singapore [5]Guangdong Institute of Intelligence Science and Technology, Zhuhai, China [6]College of Computer Science and Technology, Zhejiang University. Correspondence to: Qi Xu <xuqi@dlut.edu.cn>.

*Proceedings of the 42nd International Conference on Machine Learning*, Vancouver, Canada. PMLR 267, 2025. Copyright 2025 by the author(s).

## 1. Introduction

In recent years, artificial neural networks (ANNs) have achieved remarkable success across a range of domains, including computer vision, natural language processing, and speech recognition (Liu et al., 2024c). However, as model complexity increases, ANNs often demand substantial memory and computational resources, which can be a significant challenge in resource-constrained environments, such as embedded systems or mobile devices (Hubara et al., 2016). In contrast, spiking neural networks (SNNs), inspired by the biological mechanisms of the brain, offer an alternative approach (Liu et al., 2025). SNNs employ an event-driven spike firing mechanism for information transmission (Stöckl & Maass, 2021), where neurons are only activated in response to incoming spikes, and only trigger spikes when their membrane potential surpasses a certain threshold (Liu et al., 2024b). Outside of these events, neurons remain dormant. This asynchronous operation results in significantly lower energy consumption compared to traditional deep ANNs (Xu et al., 2024b). Furthermore, SNNs primarily rely on accumulation operations, which are less computationally expensive than the multiplication-accumulation operations commonly used in ANNs (Xu et al., 2023). These characteristics make SNNs particularly advantageous for hardware implementation, offering promising potential for the development of low-power, real-time information processing systems (Stöckl & Maass, 2021).

Despite the advantages of SNNs in terms of energy efficiency, their spike train output introduces a non-differentiable nature, which obstructs the application of backpropagation like ANNs (Xu et al., 2024a). To address this limitation, two main strategies have emerged for training deep SNNs. The first approach, direct training, uses surrogate gradients to enable backpropagation during layer-to-layer training. However, a significant performance gap still exists between ANNs and SNNs on complex tasks (Shen et al., 2025). To bridge this gap, efficient ANN-to-SNN conversion methods have been proposed to preserve the performance of ANNs during the conversion process, by adjusting network structures, optimizing the spike neuron firing modes, and designing novel spike encoding techniques (Pfeiffer & Pfeil, 2018). However, current ANN-to-SNN conversion typically requires long inference times to match the accuracy of the original ANN (Sengupta et al., 2019).

This is primarily due to the reliance on the equivalence between the ReLU activation function in ANNs and the firing rate of the integrate-and-fire (IF) model in SNNs (Cao et al., 2015). Therefore, although extending inference time can reduce conversion errors, excessively long inference times hinder the practical deployment of SNNs on neuromorphic hardware (Liu et al., 2024a). Another major challenge in ANN-to-SNN conversion lies in the remaining membrane potential in neurons, which is difficult to eliminate within a few time steps (Shen et al., 2024). Although various optimization techniques have been proposed, such as weight normalization (Diehl et al., 2015), threshold readjustment(Sengupta et al., 2019), soft reset (Han & Roy, 2020), and threshold offset (Deng & Gu, 2021), these methods often require tens to hundreds of time steps to achieve conversion accuracy comparable to the original ANN.

In order to obtain high-performance SNNs with ultra-low latency (e.g., 2 time steps), we construct an ANN-to-SNN equivalent mapping model, analyze the errors in the conversion process, and provide solutions to reduce the errors in the conversion process. Our main contributions are summarized as follows:

- This paper first analyzes the three types of errors in the ANN-to-SNN conversion process and proposes a comprehensive training framework to address these issues. The framework includes: replacing the ReLU activation function with a clipping function to train the source ANNs, mapping the weights and threshold parameters to facilitate ANN-to-SNN conversion, and setting the optimal initial membrane potential for each layer of spiking neurons.

- We introduce a clipping function with a learnable threshold, a dual-threshold neuron design, and a membrane potential initialization strategy, to jointly address the three types of errors in the conversion. Specifically, the clipping error is mitigated by mapping the threshold learned during ANN training. The dual-threshold mechanism addresses the imbalance in membrane potential activation, where excessive activation of the positive potential is often neglected, and the negative potential is underutilized. Finally, the quantization error is reduced by appropriately initializing the membrane potential, leading to a high-precision, low-latency SNN.

- We evaluate the proposed ANN-to-SNN conversion method on the CIFAR-10, CIFAR-100 and ImageNet datasets. Compared with other existing methods based on ANN-to-SNN conversion and methods that directly train through surrogate gradients, our method achieves higher accuracy in fewer time steps. For example, our method achieves 94.75% accuracy on the CIFAR-10 dataset in only 2 time steps under the ResNet-18 structure, which is significantly better than others.

## 2. RELATED WORK

The research on ANN-to-SNN conversion began with Cao et al. (2015), and then Diehl et al. (2015) successfully converted a three-layer convolutional neural network (CNN) to a spiking neural network (SNN) through data- and model-based normalization methods. In order to improve the performance of SNNs on complex datasets and deeper networks, Rueckauer et al. (2016) and Sengupta et al. (2019) proposed more precise scaling strategies for normalizing network weights and adjusting thresholds, respectively, which were later shown to be equivalent (Ding et al., 2021). However, due to the conversion error discussed in Chapter 3.1, deep SNNs often require hundreds of time steps after conversion to achieve accurate results. To address the potential information loss problem, Rueckauer et al.(2016) and Han et al.(2020) proposed an alternative to "reset-by-subtraction" neurons to replace the "reset-to-zero" neurons. In recent years, methods to eliminate conversion errors have emerged in an endless stream. Rueckauer et al. (2016) suggested using the 99.9% activation percentile as a scaling factor, while Ho & Chang (2021) introduced trainable cropping layers. In addition, Han et al. (2020) prevented improper activation of spiking neurons by rescaling the SNN threshold.

Our work has similarities with the work of Deng & Gu (2021) and Li et al. (2021), which also revealed errors in the conversion process. Deng & Gu (2021) reduced the layer error by introducing additional biases after SNN conversion, while Li et al. (2021) further proposed a method to calibrate weights and biases through quantization fine-tuning. They successfully achieved good results in the case of 16 and 32 time steps without using excessively long time steps. Bu et al. (2023) proposed to use of a quantization clip-floor-shift activation function to train ANNs, which can only minimize some of the conversion errors. In contrast, our research goal is to eliminate these conversion errors by combining existing technologies and achieving an effective transition from ANNs to SNNs. In addition, we also improve the overall performance by introducing quantized layers trained end-to-end, so that SNNs can perform well at ultra-low time steps and longer time steps. Even supervised learning methods such as backpropagation through time (BPTT) are challenging to maintain SNN performance in fewer time steps. Although BPTT methods can reduce the number of time steps required with sufficient training, they are computationally expensive, especially when training on GPUs (Neftci et al. (2019); Lee et al. (2020); Zeng and Vogels, (2021)). The timing-based backpropagation methods (Tavanaei et al. (2019); Kim et al. (2020)) can train SNNs in shorter time windows, typically only 5-10 time steps,

but these methods are generally limited to processing simpler datasets such as MNIST (Kheradpisheh & Masquelier, 2020) and CIFAR10 (Zhang & Li, 2020). In addition, Rathi et al. (2020) proposed to initialize SNNs through conversion methods and further adjust them using STDP to shorten the simulation time step. In this paper, our proposed method achieves high-performance SNNs running in only 2 time steps, showing the advantage of ultra-low latency.

## 3. Efficient ANN-SNN Conversion with Error Compensation Learning

### 3.1. Conversion Error Analysis

In this section, the conversion error between the source ANN and the converted SNN in each layer is analyzed in detail. The conversion error will lead to performance degradation in the conversion process, especially in the case of low latency. Although the error can be significantly mitigated by increasing the number of discrete states (higher inference latency), it will be detrimental to the computational efficiency and energy efficiency of real-time applications. The growing inference latency will proportionally increase the number of operations and actual running time of the deployed SNN, which offsets the advantages of deep spiking neural networks, so it is particularly important to deeply study the error in the ANN-to-SNN conversion process. The conversion error comes from the activation deviation, that is, the deviation between the ANN activation value and the SNN spike firing rate. As analyzed in the literature(Bu et al., 2023), the conversion error in the ANN-to-SNN conversion process can be specifically divided into three categories: clipping error, quantization error, and uneven error. Figure 1a shows the clipping error and quantization error, and Figure 1b- 1d shows the unevenness error.

In the conversion process from ANN to SNN, clipping error refers to the error that occurs when mapping continuous-valued weights (usually used in ANN) to discrete weights in SNN. In ANN, weights are continuous and can take any real value. However, in SNN, since the activation of neurons is represented by spike transmission, weights are usually discretized into a series of integers or binary forms. Clipping error occurs when continuous weights are mapped to discrete weights in SNN. Since the weights in SNN are discrete, the continuous weights in ANN cannot be accurately represented, which will lead to some information loss and errors. Clipping errors will have a negative impact on the performance of SNN. Especially in tasks with high precision requirements, clipping errors will cause the performance of SNN to degrade. Therefore, when converting ANN to SNN, it is necessary to consider how to minimize clipping error, and some adjustments and parameter optimization are required.

The output $\phi^l(T)$ of the SNN is in the range $[0, \theta^l]$. However, the output $a^l$ of the ANN is in the larger range $[0, a^l_{max}]$, where $a^l_{max}$ represents the maximum value of $a^l$. $a^l$ can be mapped to $\phi^l(T)$ by the following equation:

$$\phi^l(T) = clip\left(\frac{\theta^l}{T}\left\lfloor\frac{a^lT}{\lambda^l}\right\rfloor, 0, \theta^l\right) \qquad (1)$$

Here the clip function sets the upper limit $\theta^l$ and the lower limit 0. $\lfloor\cdot\rfloor$ indicates floor function. $\lambda^l$ represents the actual maximum value of the output $a^l$ mapped to the maximum value $\theta^l$ of $\phi^l(T)$. Rueckauer et al.(2017) concluded that 99% of the activation values in ANN are in the range $[0, \frac{a^l_{max}}{3}]$, so the threshold $\lambda^l$ is set to 99% of the maximum activation value to avoid the influence of outliers and thus reduce the inference delay in the conversion. The activation between $\lambda^l$ and $a^l_{max}$ in the ANN is mapped to the same value $\theta^l$ in the SNN, which will cause a conversion error called clipping error. In Figure 1a, the blue line represents the ReLU activation function, the red line represents the IF neuron, and the dotted line position pointed to by the clipping error shows the part where the maximum activation value of the ANN is greater than the maximum spike firing rate of the SNN. This part of the activation value cannot be accurately represented by the spike firing rate, so it can only be represented by the maximum spike firing rate, resulting in clipping error.

Quantization errors arise from emulating continuous ANN activation values with discrete SNN firing rates. The spike rate output value $\phi^l(T)$ of the SNN is discrete, resulting in a quantization resolution $\frac{\theta^l}{T}$, As shown in Figure 1a, $z^l \in \left[\frac{k\lambda^l}{T}, \frac{(2k+1)\lambda^l}{2T}\right], (k = 0, 1, ..., T)$ is mapped to $\phi^l(T)$ and will be rounded down to $\frac{k\lambda^l}{T}$, $z^l \in \left[\frac{(2k-1)\lambda^l}{2T}, \frac{k\lambda^l}{T},\right], (k = 0, 1, ..., T)$ will be rounded up to $\frac{\theta^l}{T}$, causing quantization error. The black dashed line at the position pointed by the quantization error represents the quantization error. At this position, the activation values of the continuous ANN cannot be accurately simulated by the spike firing rate of the SNN, but are represented by the same spike firing rate value.

The unevenness error originates from the non-uniform distribution of the input spike arrival time in the total time step. The discussion of the previous two types of errors is based on the assumption that the input spike arrival time is uniformly distributed. However, the spikes received by the deep layer are often non-uniform to some extent, which leads to differences in the output of the layer, and then to non-uniform errors. As shown in Figure 1b, 1c and 1d, the three cases show that non-uniform inputs can lead to more or fewer output spikes compared to the ideal case. Ideally, as shown in Figure 1b, assume that the two simulated neu-

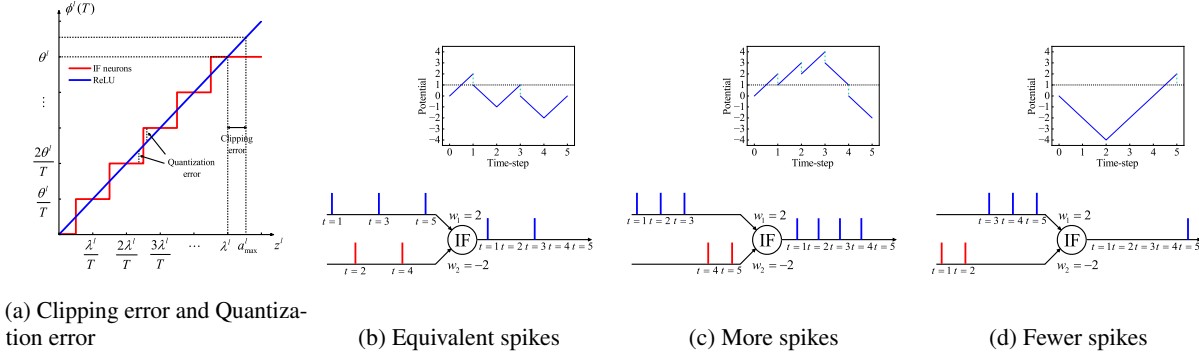

(a) Clipping error and Quantization error

(b) Equivalent spikes

(c) More spikes

(d) Fewer spikes

*Figure 1.* Three types of errors in ANN-SNN conversion

rons in the $l-1$ layer are connected to a simulated neuron in the $l$ layer, whose weights $w^l = [2, -2]$, and the output vector $a^{l-1}$ of the neuron in the $l-1$ layer is $[0.6, 0.4]$, the threshold $\theta^{l-1} = 1$, $T = 5$, and the ANN activation value $a^l = \frac{2}{5}$ is calculated. Then, the average postsynaptic membrane potential is calculated by the internal membrane potential of the IF neuron. As shown in the box in the figure, the internal membrane potential change diagram shows that the spiking neuron emits spikes at $t = 1$ and $t = 3$, and $\phi^l(T) = \frac{\sum_{i=1}^{T} s^l(i)\theta^l}{T} = \frac{2}{5}$ is calculated, which is the same as $a^l$. Therefore, the activation value of the $l$ layer of the ANN is correctly simulated. However, in the cases of Figure 1c and 1d, the spike train output generated according to the membrane potential change in the box is calculated as $\phi^l(T) = \frac{4}{5}$ and $\phi^l(T) = \frac{1}{5}$, respectively, which are not equal to $a^l$ in terms of value. These three cases can intuitively demonstrate the unevenness error.

There are some limitations in the existing ANN-SNN conversion methods. In previous studies, these three errors can be reduced by replacing the activation function in the neural network. Specifically, Opt (Bu et al., 2022) uses the settings of learnable upper bounds and membrane potential initialization to eliminate the clipping error and reduce the quantization error to a certain extent. Quantized activation functions are proposed to replace the ReLU activation function in ANN, thereby reducing the quantization error in the conversion process (Yan et al., 2021). SRP (Hao et al., 2023) proposed an optimal strategy based on residual membrane potential to eliminate uneven errors. However, how to solve these three types of conversions at the same time remains a challenge. For image classification tasks, a pre-trained model is usually required to fine-tune the network. However, using a pre-trained model with ReLU to fine-tune a model with a quantization function is not ideal. It is meaningless to use an ANN network with unsatisfactory results to convert it into an SNN. Therefore, the purpose of this paper is to solve the three conversion errors in ANN-SNN conversion in image classification tasks and achieve high performance

of the model with extremely low latency.

### 3.2. Clipping Function with Learnable Threshold

In ChapterA.2, we analyzed the principle of the ANN-SNN conversion method. We hope to directly train the appropriate threshold value of each layer through ANN to realize threshold mapping, and then directly map the ANN weight to SNN to realize the conversion from ANN to SNN. Specifically, first, in order to reduce the clipping error, we use a clipping function with a learnable threshold $\lambda$ to replace the ReLU activation function in the ANN. We use the quantization clip-floor-shift activation function proposed by (Bu et al., 2023) to train the ANN, and the formula is as follows:

$$a^l = \hat{h}(z^l) = \lambda^l clip\left(\frac{1}{L}\lfloor \frac{z^l L}{\lambda^l} + \varphi \rfloor, 0, 1\right) \quad (2)$$

The hyperparameter $L$ represents the quantization step size of the ANN, and the displacement term $\varphi$ is set to $\frac{1}{2}$. Note that $z^l = W^l\phi^{l-1} = W^la^{l-1}$. Then, train the ANN with the clipping function. This learnable threshold $\lambda$ is obtained through training in the ANN and can be directly mapped to the corresponding layer of the SNN. Finally, the learned threshold $\lambda$ in the ANN is directly mapped to the spike emission threshold $\theta$ in the SNN, and the weight parameters of the ANN are also mapped to the weight parameters of the SNN to complete the conversion from ANN to SNN. Since in the threshold mapping process, the learnable threshold $\lambda$ in the ANN is equal to the spike emission threshold $\theta$ in the SNN, the numerical difference between the two thresholds disappears, so the clipping error can be avoided.

### 3.3. Dual Threshold Neuron

In response to the three types of errors analyzed in this chapter, this section proposes a dual-threshold neuron that effectively alleviates the quantization error and unevenness error in the conversion, while eliminating the clipping error

caused by the scaling threshold. Specifically, due to the observation of the changes in the membrane potential inside the IF neuron, it was found that the positive membrane potential is always over-released, while the release of the negative membrane potential is always ignored. This was ignored in previous work, according to Equation 3, because when the time step $T$ is long enough (that is, hundreds or thousands of time steps), it has little effect on the performance.

$$\frac{N_i^l}{T} \approx \frac{N_i^l + 1}{T} \tag{3}$$

where $N_i^l$ is the number of output spikes. However, when reducing the delay $T$ to a few time steps, the unevenness error will significantly affect the approximation between the ANN activations and the SNN firing rate. An example of the cause and effect of the unevenness error and the principle of the dual threshold neuron to solve the unevenness error is demonstrated in Figure 2.

In the dual-threshold neuron model, a neuron can only fire a negative spike if it reaches the negative discharge threshold and has fired at least one positive spike. In order to restore the incorrectly subtracted membrane potential, our model changes the reset mechanism of negative spikes by adding a positive threshold (i.e., $\theta$). The spike function $\Theta(t)$ is then rewritten as:

$$\Theta(t) = \begin{cases} 1, & \text{if } V_i^l(t) > \theta, \\ -1, & \text{if } V_i^l(t) \le \theta' \text{ and } N_i^l(t) \ge 1, \\ 0, & \text{otherwise, no firing.} \end{cases} \tag{4}$$

Where $\theta'$ is the negative threshold and $N_i^l(t)$ is the number of spikes fired by neuron $i$ at time step $t$. In order to improve the sensitivity of negative spikes, this paper sets the negative threshold to a small negative value (-1e-3 according to experience). Then the membrane dynamics update rule of the IF neuron is rewritten as:

$$\begin{aligned} V_i^l(t) =& V_i^l(t-1) + z_i^l(t) - \theta^l \Theta(V_i^l(t) - \theta^l) \\ &+ \theta^l \Theta(\theta' - V_i^l(t)) \Theta(N_i^l - 1) \end{aligned} \tag{5}$$

According to Equation 4 and Equation 5, the IF neuron will fire a negative spike to cancel the erroneously fired spike and restore the membrane potential.

### 3.4. Initialization of Membrane Potential

The complete equivalence between the forwarding process of the ANN and the spike firing rate (or postsynaptic potential) of the adjacent layers of the SNN introduced in Chapter A.2 depends on the simulation time step $T$ being

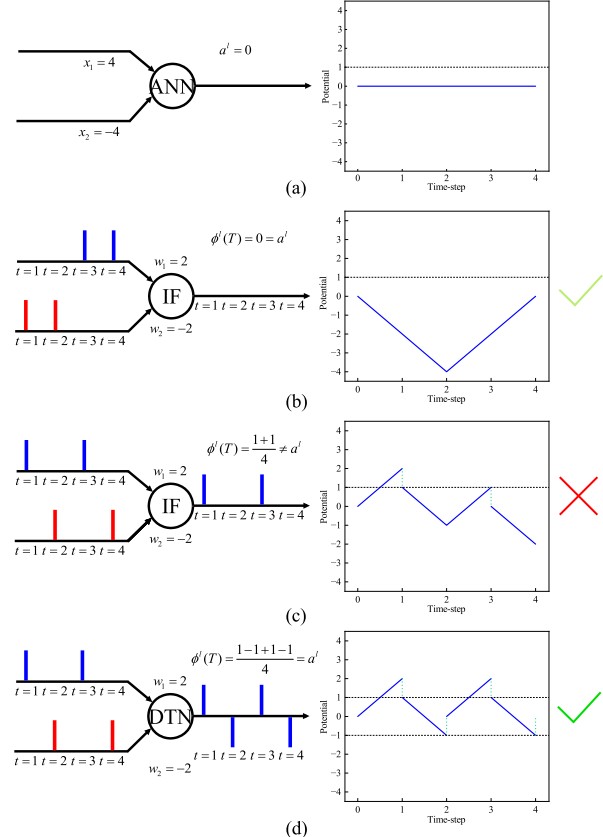

*Figure 2.* The principle of dual threshold neuron. (a) The ANN neuron receives two inputs: 4 and -4, and its output is 0; For SNN, assuming the threshold $\theta^l = 1$. (b) The IF neuron receives four spike charges $(-2, -2, 2, 2)$ at $t = 1, 2, 3, 4$. Its output firing rate is equivalent to ANN activation. (c) The IF neuron receives four spike charges $(2, -2, 2, -2)$ at $t = 1, 2, 3, 4$, However it will immediately fire a spike at $t = 1$ because the membrane potential is greater than the excitation threshold. No spike is fired at $t = 2$. When $t = 3$, another spike is fired because the membrane potential is greater than the excitation threshold. When $t = 4$, no spike is fired, resulting in a firing rate that is not equal to the ANN output activation. (d) DTN represents the dual threshold neuron proposed in this paper. The dual threshold neuron receives three spike charges $(2, -2, 2, -2)$ at $t = 1, 2, 3, 4$. Although it fires a spike at $t = 1$, the dual threshold neuron model of this paper outputs a negative spike at $t = 2$. When $t = 3$, the neuron fires a spike. The dual threshold neuron model of this paper outputs a negative spike at $t = 4$, which offsets the residual membrane potential, thereby generating a firing rate equivalent to that emitted by the ANN.

large enough so that $\frac{v^l(t)}{T} \approx 0$, $\frac{v^l(0)}{T} \approx 0$, which results in a long inference time for the SNN when applied to complex data sets. In addition, under the constraint of low latency, there is an inherent difference between the activation value of the ANN and the average postsynaptic potential of the SNN, which is propagated layer by layer, resulting in a significant decrease in the accuracy of the converted SNN. This section will analyze the impact of membrane potential initialization and show that the optimal initialization can achieve the expected error-free ANN to SNN conversion. Since $\phi^l(T) \geq 0$, the equivalent form of Equation 17 is as follows:

$$\phi^l(T) = \max\left(w^l \cdot \phi^{l-1}(T) - \frac{v^l(T) - v^l(0)}{T}, 0\right) \quad (6)$$

According to the above formula, the relationship between the average postsynaptic potentials of adjacent layer spike neurons can be rewritten in a new way:

$$\phi^l(T) = \max\left(\frac{\theta^l}{T}\left\lfloor\frac{T \cdot w^l \cdot \phi^{l-1}(T) + v^l(0)}{\theta^l}, 0\right\rfloor\right) \quad (7)$$

From the above formula, we can see that the spike firing rate is discrete, and there is an inherent quantization error between ANN and SNN. This section studies an optimal membrane potential initialization method to reduce the conversion error. Specifically, assume that the ReLU activation of the $l$ layer in the ANN is the same as the average postsynaptic membrane potential of the $l$ layer in the SNN, and then compare the outputs of the $l$ layer of the ANN and SNN. For convenience, $f(z) = f(W^l \cdot a^{l-1}) = \max(W^l \cdot a^{l-1}, 0)$ is used to represent the output value of the ANN, and $\hat{f}(z) = f(W^l \cdot \phi^{l-1}(T)) = \max\left(\frac{\theta^l}{T}\left\lfloor\frac{T \cdot z + v^l(0)}{\theta^l}, 0\right\rfloor\right)$ is used to represent the output value of the SNN. The expected square difference between $a^l$ and $\phi^l(T)$ can be defined as:

$$E_z\left\|f(z) - \hat{f}(z)\right\|_2^2 = E_z\left\|z - \frac{\theta^l}{T}\left\lfloor\frac{T \cdot z + v^l(0)}{\theta^l}\right\rfloor\right\|_2^2 \quad (8)$$

Through mathematical derivation, it is found that $\arg\min_{v^l(0)} E_z\left\|f(z) - \hat{f}(z)\right\|_2^2 = \frac{\theta^l}{2}$, when the initial membrane potential is set to half the threshold, the expected value of the square conversion error reaches the minimum, thereby minimizing the conversion error.

## 4. Experiments

To evaluate the effectiveness of the ANN-SNN conversion method proposed in this paper, image classification was taken as the target task, and its performance was evaluated on the CIFAR-10, CIFAR-100 and ImageNet datasets to validate the effectiveness of our method. Similar to previous work, we use VGG-16, ResNet-18, and ResNet-20 network structures as the source ANN. And compare the method proposed in this article with other state-of-the-art methods, including RMP from Han et al.(2020), TSC from Han & Roy(2020), ReLUThreshold-Shift (RTS) from Deng & Gu,(2021), RNL from Ding et al.(2021), Opt from Bu et al.(2022), Quantization Clip-Floor-Shift activation function (QCFS) from Bu et al.(2023), and SNN Conversion with AdvancedPipeline (SNNC-AP) from Li et al.(2021).

### 4.1. Comparison with state-of-the-art Methods

Table 1 shows the performance comparison between the proposed conversion method and the state-of-the-art ANN-SNN conversion method on CIFAR-10. Among them, the accuracy of the comparison methods is based on the experimental data in the original paper, and the best accuracy is highlighted in bold. On the CIFAR-10 dataset, regardless of whether the model is VGG-16, ResNet-18, or ResNet-20, our method outperforms previous conversion methods at low time steps. Specifically, on VGG-16, the accuracy of our method reached 95.15% at $T = 4$, which is comparable to the performance of ANN. On ResNet-20, the accuracy of our method reached 90.55% at $T = 4$, which is similar to the performance of ANN. On ResNet-18, the accuracy of our method reached 95.99% at $T = 4$, which is also similar to the performance of ANN. These results validate the effectiveness of the method proposed in this paper. It is worth noting that when $T = 2$, the accuracy of our method on ResNet-18 reaches 94.75%, approaching the accuracy of ANN in a very short time step.

We also evaluate the performance of our method on the large-scale dataset ImageNet and list the results in Table 2. Our method also outperforms other methods in terms of high accuracy and ultra-low latency. For VGG-16, when T=16, our method achieves 74.15% accuracy, which is 23.18% higher than QCFS. For ResNet-34, we achieve 72.37% accuracy with only 16 time steps. These results show that our method outperforms previous conversion methods.

### 4.2. Effect Evaluation of Dual Threshold Neuron

To verify the effectiveness of the dual threshold neuron model proposed in this paper, this section gradually applies each part of our method to the transformed SNN. Train VGG-16, ResNet-20, and ResNet-18 using the CIFAR-10 and CIFAR-100 datasets, and then convert the trained ANN to SNN. Here, this section introduces a set of SNN concepts. SNN$\alpha$ represents the native SNN directly converted from the trained ANN, while SNN$\beta$ represents the SNN$\alpha$ using dual threshold neuron. As shown in Table 3 and Table 5, the

*Table 1.* Comparison of the proposed method with other methods on the CIFAR-10 dataset

| Architecture | Method | ANN | T=2 | T=4 | T=8 | T=16 | T=32 | T=64 |
|---|---|---|---|---|---|---|---|---|
| | RMP | 93.63% | - | - | - | - | 60.30% | 90.35% |
| | TSC | 93.63% | - | - | - | - | - | 92.79% |
| | RTS | 95.72% | - | - | - | - | 76.24% | 90.64% |
| VGG-16 | RNL | 92.82% | - | - | - | 57.90% | 85.40% | 91.15% |
| | Opt | 94.57% | - | - | 90.96% | 93.38% | 94.20% | 94.45% |
| | QCFS | 95.52% | 91.18% | 93.96% | 94.95% | 95.40% | 95.54% | 95.55% |
| | **Ours** | **95.23%** | **93.53%** | **95.15%** | **95.41%** | **95.39%** | **95.48%** | **95.43%** |
| | TSC | 91.47% | - | - | - | - | - | 69.38% |
| ResNet-20 | QCFS | 91.77% | 73.20% | 83.75% | 89.55% | 91.62% | 92.24% | 92.35% |
| | **Ours** | **91.76%** | **84.52%** | **90.55%** | **92.10%** | **92.40%** | **92.43%** | **92.46%** |
| | RTS | 95.46% | - | - | - | - | 84.06% | 92.48% |
| | RNL | 93.84% | - | - | - | 47.63% | 83.95% | 91.96% |
| ResNet-18 | Opt | 96.04% | - | - | 75.44% | 90.43% | 94.82% | 95.92% |
| | QCFS | 95.04% | 75.44% | 90.43% | 94.82% | 95.92% | 96.08% | 96.06% |
| | **Ours** | **96.39%** | **94.75%** | **95.99%** | **96.11%** | **96.40%** | **96.35%** | **96.39%** |

*Table 2.* Comparison of the proposed method with other methods on the ImageNet dataset

| Architacture | Method | ANN | T=16 | T=32 | T=64 | T=128 | T=256 |
|---|---|---|---|---|---|---|---|
| | TSC | 70.64% | - | - | - | - | 61.48% |
| ResNet-34 | RTS | 75.66% | - | 0.09% | 0.12% | 3.19% | 47.11% |
| | SNNC-AP | 75.66% | - | 64.54% | 71.12% | 73.45% | 74.61% |
| | QCFS | 74.32% | 59.35% | 69.37% | 72.35% | 73.15% | 73.37% |
| | **Ours** | **74.36%** | **72.37%** | **73.16%** | **73.31%** | **73.38%** | **-** |
| | RMP | 73.49% | - | - | - | - | 48.32% |
| VGG-16 | TSC | 73.49% | - | - | - | - | 69.71% |
| | SNNC-AP | 75.36% | - | 63.64% | 70.69% | 73.32% | 74.23% |
| | QCFS | 74.29% | 50.97% | 68.47% | 72.85% | 73.97% | 74.22% |
| | **Ours** | **73.85%** | **74.15%** | **74.29%** | **74.31%** | **74.32%** | **-** |

dual threshold neuron model improved the performance of the transformed SNN.

*Table 3.* Comparison of accuracy between native SNN and SNN using dual threshold neuron model on CIFAR-10 dataset

| Dataset | T | Architecture | SNN$\alpha$ | SNN$\beta$ | $\Delta$Acc |
|---|---|---|---|---|---|
| | | VGG-16 | 91.18% | 93.53% | +2.35% |
| | 2 | ResNet-20 | 73.20% | 84.52% | +11.32% |
| | | ResNet-18 | 75.44% | 94.75% | +19.31% |
| | | VGG-16 | 93.96% | 95.15% | +1.19% |
| CIFAR-10 | 4 | ResNet-20 | 83.75% | 90.55% | +6.80% |
| | | ResNet-18 | 90.43% | 95.99% | +5.56% |
| | | VGG-16 | 94.95% | 95.41% | +0.46% |
| | 8 | ResNet-20 | 89.55% | 92.10% | +2.55% |
| | | ResNet-18 | 94.82% | 96.11% | +1.29% |

The dual threshold neuron model in this article continuously improves the performance of the converted SNN, as shown in Figure 3. On the CIFAR-10 dataset, the SNN$\beta$ with the dual threshold neuron model can improve the performance of SNN$\alpha$. The improvement of ResNet-18 at $T = 2$ reaches 19.31%. It is worth noting that the method proposed in this article achieved comparable accuracy to the source ANN network on the CIFAR-10 dataset at $T = 4$, and even at a time step of $T = 2$, the converted SNN achieved high accuracy. More information on the CIFAR-100 improvement is

in Figure 5.

### 4.3. Effect of Quantization Steps $L$

In our dual threshold neuron method, the quantization steps $L$ is a hyperparameter that affects the accuracy of the converted SNN. To better understand the impact of $L$ on SNN performance and determine the optimal value, we trained VGG16, ResNet-20, and ResNet-18 networks with a pruning function with a learnable threshold $\lambda$ using different quantization steps $L$, including 2, 4, 8, 16, and 32, and then converted them to SNNs. The results in Table 7 and Figure 4, Figure 6 illustrate the impact on the accuracy of the converted SNNs under different quantization steps $L$. We found that the accuracy of SNNs at ultra-low latency (within 4 time steps) decreases when the quantization steps $L$ increases. However, too small a quantization step (e.g., 2 time steps) reduces the model capacity and leads to reduced SNN accuracy. The recommended quantization step $L$ is 4 or 8, which can achieve high-performance converted SNNs at small time steps and very large time steps.

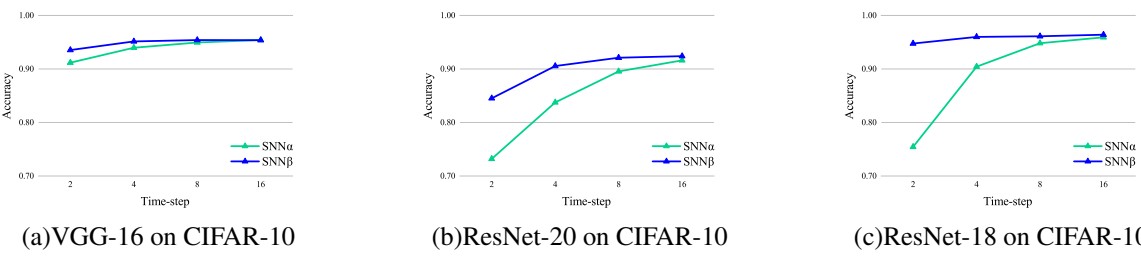

(a)VGG-16 on CIFAR-10      (b)ResNet-20 on CIFAR-10      (c)ResNet-18 on CIFAR-10

*Figure 3.* Evaluation of the dual threshold neuron model on CIFAR-10

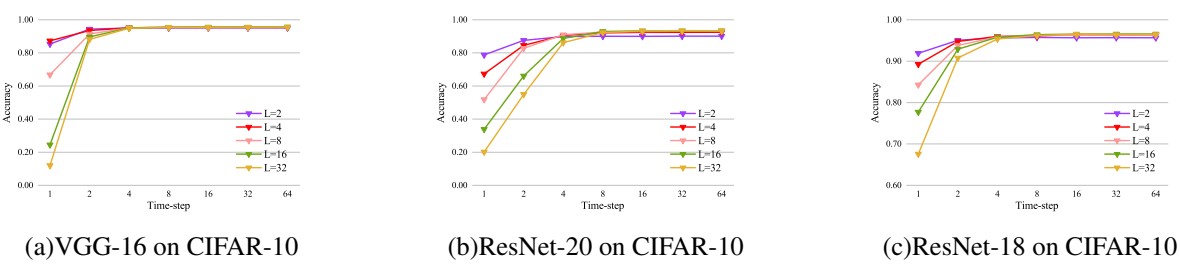

(a)VGG-16 on CIFAR-10      (b)ResNet-20 on CIFAR-10      (c)ResNet-18 on CIFAR-10

*Figure 4.* Influence of different quantization steps $L$ on CIFAR-10

## 4.4. Energy Consumption Analysis

On the principle level, SNNs have attracted much attention due to their unique properties. The neurons in SNN only perform computational operations when spikes occur, resulting in power consumption. In contrast, without spike activation, these neurons do not perform any calculations and therefore do not generate additional power consumption. This low-power characteristic gives SNN significant advantages in hardware deployment, especially in resource constrained environments. In addition, the power consumption of SNN is positively correlated with neuronal spike activity, which provides the possibility for energy-efficient computing modes. This means that SNNs can complete complex computing tasks with lower energy consumption and adjust spike emission strategies according to application requirements during design to achieve better energy efficiency ratios. Therefore, the advantages of SNN in power management bring potential benefits for its future technological development and application deployment.

At the experimental level, in order to further investigate the power consumption of spike neural networks, this section evaluated the energy consumption of the method used in this paper on the CIFAR-100 dataset. This section uses the VGG-16 network structure. According to the analysis, this article uses the synaptic operation (SOP) of SNN to represent the basic number of operations required for classifying an image. Use 77fJ/SOP for SNN and 12.5pJ/FLOP as the power consumption benchmark for ANN (Qiao et al., 2015). The power consumption analysis is shown in Table 4.

It is worth noting that the SNN converted using the method

*Table 4.* Energy consumption analysis using VGG-16 on CIFAR-100 dataset

| | ANN | T=2 | T=4 | T=8 | T=16 | T=32 |
|---|---|---|---|---|---|---|
| Accuracy | 76.46% | 72.58% | 76.50% | 77.14% | 77.29% | 77.39% |
| Energy(mJ) | 4.170 | 0.004 | 0.008 | 0.017 | 0.034 | 0.068 |

proposed in this article achieved an accuracy of 77.29% at $T = 16$, which is very close to the accuracy of the source ANN and only requires 0.034mJ of energy cost, significantly lower than the energy consumption of 4.170mJ of the source ANN. The method proposed in this article achieved an accuracy of 76.50% for SNN at $T = 4$ and only requires 0.008mJ of energy cost.

The SNN converted using the method proposed in this article exhibits excellent energy efficiency. These results clearly demonstrate the high inference performance and energy efficiency achieved by the method proposed in this article. The comparison of accuracy and power consumption highlights its effectiveness, allowing the converted SNN to operate in a highly energy-efficient manner while achieving excellent performance.

## 5. Conclusion

In this paper, we propose an ANN-to-SNN conversion method to achieve high-precision and ultra-low latency deep SNNs. Specifically, we use a new activation function to replace the conventional ReLU activation function in ANN, which is closer to the activation characteristics of SNN while keeping the ANN performance almost unchanged. In ad-

dition, we propose a dual-threshold neuron that effectively alleviates the quantization error and uneven error that appear in the conversion, while eliminating the clipping error that is re-generated by scaling the threshold. We achieve state-of-the-art accuracy on CIFAR-10, CIFAR-100 and ImageNet datasets with fewer time steps. Our research results not only promote the application of neuromorphic hardware, but also lay the foundation for the large-scale practical deployment of SNNs.

## Acknowlegdements

This work was supported by the National Natural Science Foundation of China under Grant (No.62476035, 62206037, 62306274, 61925603, U24B20140), and in part by the Young Elite Scientists Sponsorship Program by CAST under Grant 2024QNRC001, Open Research Program of the National Key Laboratory of Brain-Machine Intelligence, Zhejiang University (No. BMI2400012).

## Impact Statement

This paper presents work whose goal is to advance the field of Deep Learning. There are many potential societal consequences of our work, none which we feel must be specifically highlighted here.

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

# A. PRELIMINARIES

## A.1. Neuron Model for SNNs

Spiking neural networks use spiking neurons as their basic computing units, which simulate the natural mechanism of biological neurons communicating through spike trains. In the field of neuroscience research, a variety of spiking neuron models have been proposed to accurately simulate the input-output relationship of biological neurons. This section will focus on two commonly used spiking neuron models: the LIF neuron model and the IF neuron model. Both neuron models are used to simplify the description of the spike emission process. The dynamic expression of the LIF neuron is:

$$\tau \frac{dV(t)}{dt} = -(V(t) - V_{rest}) + X(t), \ \ V(t) < V_{th} \tag{9}$$

Among them, $V(t)$ represents the membrane potential of the IF neuron at time $t$, $X(t)$ represents the input of the neuron at time $t$, $\tau$ is the membrane time constant, and $V_{rest}$ is the resting potential. When the membrane potential $V(t)$ exceeds the threshold $V_{th}$ at time $t$, the neuron fires a spike, and then the membrane potential $V(t)$ returns to the resting potential $V_{rest}$. In practical applications, the dynamics need to be discretized. The discretized LIF model is described as:

$$
\begin{aligned}
m^l(t) &= \tau \cdot v^l(t-1) + W^l \cdot x^{l-1}(t), \\
s^l(t) &= H(m^l(t) - \theta^l), \\
v^l(t) &= m^l(t) - s^l(t) \cdot \theta^l.
\end{aligned}
\tag{10}
$$

Among them, $m^l(t)$ represents the membrane potential of the neuron before the trigger spike at the $t$ time step, $v^l(t)$ represents the membrane potential of the neuron after the trigger spike at the $t$ time step, and $W^l$ represents the weight matrix between the $l$ layer and the $l-1$ layer, $x^{l-1}(t)$ is the IF neuron in the $l$ layer receiving input from the last layer. At the same time, the firing threshold in the $l$ layer LIF model is $\theta^l$, $H(\cdot)$ is the Heaviside step function, and $s^l(t)$ represents the spike output of the $l$ layer neuron at time $t$, the element of which equals 1 if there is a spike and 0 otherwise. For each element in $m^l(t)$, if $m_i^l(t) > \theta^l$, the neuron will fire a spike and update the membrane potential $v_i^l(t)$. In order to avoid information loss, a soft reset mechanism is usually used to update $v^l(t)$ instead of directly resetting it to 0, that is, when a spike is triggered, the membrane potential $v^l(t)$ minus the threshold $\theta^l$. In the discretized representation, the resting potential is usually set to 0, so the threshold $\theta^l > 0$. LIF neurons strike a balance between computational cost and biological plausibility.

The dynamic expression of IF neurons is:

$$\frac{dV(t)}{dt} = -(V(t) - V_{rest}) + X(t), \ \ V(t) < V_{th} \tag{11}$$

Compared with the LIF model, the IF model simplifies the membrane time constant term, that is, the leakage of membrane potential over time is ignored. The discretization description of the IF model is:

$$
\begin{aligned}
m^l(t) &= v^l(t-1) + W^l \cdot x^{l-1}(t), \\
s^l(t) &= H(m^l(t) - \theta^l), \\
v^l(t) &= m^l(t) - s^l(t) \cdot \theta^l.
\end{aligned}
\tag{12}
$$

This paper uses the IF model for ANN-SNN conversion, which allows neurons to accumulate potential through input current and emit spikes when the threshold is reached. Compared with the LIF model, the IF model does not contain leakage terms, so it is more suitable for converting ANN neurons with rectified linear unit (ReLU) activation function into SNN neurons, thereby realizing ANN-SNN conversion.

## A.2. Principle of ANN-SNN Conversion Method

First, let's look at the relationship between ANN and SNN. Consider a typical ANN with $M$ fully connected layers. The output of the $l$ convolutional layer can be expressed as:

$$a^l = h(Wa^{l-1}), \ \ l = 1, 2, ..., M \tag{13}$$

Among them, the vector $a^l$ represents the output of all neurons in the $l$ layer, $W^l$ represents the weight matrix between the $l$ layer and the $l-1$ layer, and $h(\cdot)$ is the ReLU activation function. ReLU is widely used in ANN because of its piecewise linear characteristics, which simplifies the weight update process during training.

For the neuron model of SNN, the IF neuron model is usually used. Since it does not have leakage terms and refractory periods, the spike firing rate of the IF neuron in a given time window and the activation value of ReLU can be used as a bridge to connect SNN and ANN. In addition, for classification tasks, the maximum activation value of all units in the final output layer is crucial to the classification result, so scaling the spike firing rate of IF neurons in each layer by a constant factor does not affect the classification result. Similar to the approach of Deng et al.(2021), we assume that if the presynaptic neuron in layer $l-1$ fires a spike, the postsynaptic neuron in layer l receives an unweighted postsynaptic potential $\theta^l$, that is:

$$x^l(t) = s^l(t)\theta^l \tag{14}$$

The key idea of ANN-SNN conversion is to map the activation values of simulated neurons in ANN to the firing rate (or average postsynaptic potential) of spiking neurons in SNN. The spatiotemporal dynamics equation of the IF neuron is shown in Equation 12, which represents the neuron charging, discharging, and resetting process. If the IF neuron in the $l$ layer receives input $x^{l-1}(t)$ from the last layer, the discrete function between the impulses in the $l$ layer and the $l-1$ layer can be obtained by superimposing the equation:

$$v^l(t) - v^l(t-1) = W^l x^{l-1}(t) - s^l(t)\theta^l \tag{15}$$

Equation 15 describes the basic function of the spike neuron used in the ANN-SNN conversion. Summing Equation 15 from time 1 to $T$ and dividing $T$ on both sides, we get:

$$\frac{v^l(T) - v^l(0)}{T} = \frac{W^l \sum_{i=1}^{T} x^{l-1}(i)}{T} - \frac{\sum_{i=1}^{T} s^l(i)\theta^l}{T} \tag{16}$$

If $\phi^{l-1}(T) = \frac{\sum_{i=1}^{T} x^{l-1}(i)}{T}$ is used to represent the average postsynaptic potential from 0 to $T$, and Equation 14 is substituted into Equation 16, we can obtain:

$$\phi^l(T) = W^l \phi^{l-1}(T) - \frac{v^l(T) - v^l(0)}{T} \tag{17}$$

Equation 17 describes the relationship between the average postsynaptic potentials of neurons in adjacent layers. Note that $\phi^l(T) \geq 0$. If the initial membrane potential $v^l(0)$ is set to zero and the residual term $\frac{v^l(T)}{T}$ is ignored when the simulation time step $T$ is long enough, the converted SNN has almost the same activation function as the source ANN (Equation 13). However, a higher time step $T$ results in longer inference latency, which hinders the practical application of SNNs. Therefore, this paper aims to achieve high-performance ANN-SNN conversion with extremely low latency.

*Table 5.* Comparison of accuracy between native SNN and SNN using dual threshold neuron model on CIFAR-100 dataset

| Dataset | T | Architecture | SNN$\alpha$ | SNN$\beta$ | $\Delta$Acc |
|---------|---|--------------|-------------|------------|-------------|
| CIFAR-100 | 2 | VGG-16 | 63.79% | 72.58% | +8.79% |
| | | ResNet-20 | 19.96% | 42.91% | +22.95% |
| | | ResNet-18 | 70.79% | 75.11% | +4.32% |
| | 4 | VGG-16 | 69.62% | 76.50% | +6.88% |
| | | ResNet-20 | 34.14% | 61.63% | +27.49% |
| | | ResNet-18 | 75.67% | 78.81% | +3.14% |
| | 8 | VGG-16 | 73.97% | 77.14% | +3.17% |
| | | ResNet-20 | 55.37% | 67.31% | +11.94% |
| | | ResNet-18 | 78.48% | 79.48% | +1.00% |

## B. Comparison on CIFAR-100 Dataset

We evaluated the performance of our method on CIFAR-100 and listed the results in Table 6. Our method also outperforms other methods in terms of high accuracy and ultra-low latency. It is worth noting that when $T = 2$, our method achieved an accuracy of 72.58% for VGG16, which is 8.79% higher than the suboptimal QCFS method. When the time step is only 4, our method achieves an accuracy of 76.50%, which is similar to the accuracy of the source ANN. The results indicate that the method proposed in this paper outperforms previous conversion methods in both accuracy and ultra-low latency.

*Table 6.* Comparison of the proposed method with other methods on the CIFAR-100 dataset

| Architacture | Method | ANN | T=2 | T=4 | T=8 | T=16 | T=32 | T=64 |
|---|---|---|---|---|---|---|---|---|
| VGG-16 | SNNC-AP | 77.89% | - | - | - | - | 73.55% | 76.64% |
| | RTS | 70.62% | - | - | - | 65.94% | 69.80% | 70.35% |
| | QCFS | 76.28% | 63.79% | 69.62% | 73.97% | 76.24% | 77.01% | 77.10% |
| | **Ours** | **76.46%** | **72.58%** | **76.50%** | **77.14%** | **77.29%** | **77.39%** | **77.28%** |
| ResNet-20 | RMP | 68.72% | - | - | - | - | 27.64% | 46.91% |
| | QCFS | 69.94% | 19.96% | 34.14% | 55.37% | 67.33% | 69.82% | 70.49% |
| | **Ours** | **68.58%** | **42.91%** | **61.63%** | **67.31%** | **68.65%** | **69.15%** | **69.36%** |
| ResNet-18 | SNNC-AP | 77.16% | - | - | - | - | 76.32% | 77.29% |
| | RTS | 67.08% | - | - | - | 63.73% | 68.40% | 69.27% |
| | QCFS | 78.80% | 70.79% | 75.67% | 78.48% | 79.48% | 79.62% | 79.54% |
| | **Ours** | **79.09%** | **75.11%** | **78.81%** | **79.48%** | **79.45%** | **79.55%** | **79.50%** |

## C. Effect Evaluation of Dual Threshold Neuron on CIFAR-100 Dataset

Table 5 reports the comparison of accuracy between native SNN and SNN using dual threshold neuron model on CIFAR-100 dataset.

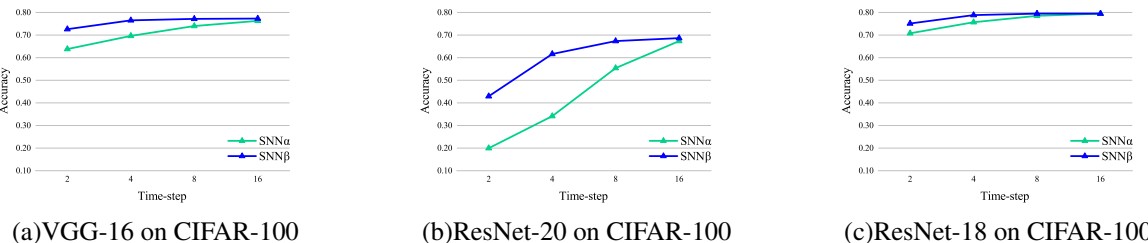

(a)VGG-16 on CIFAR-100    (b)ResNet-20 on CIFAR-100    (c)ResNet-18 on CIFAR-100

*Figure 5.* Evaluation of the dual threshold neuron model on CIFAR-100

## D. Effect of Quantization Steps $L$

Table 7 summarizes the performance of converted SNNs with varying quantization steps $L$ and time-steps $T$. For VGG-16 and quantization steps $L = 2$, we achieve an accuracy of 94.35% on CIFAR-10 dataset and an accuracy of 73.78% on CIFAR-100 dataset with 1 time-steps.

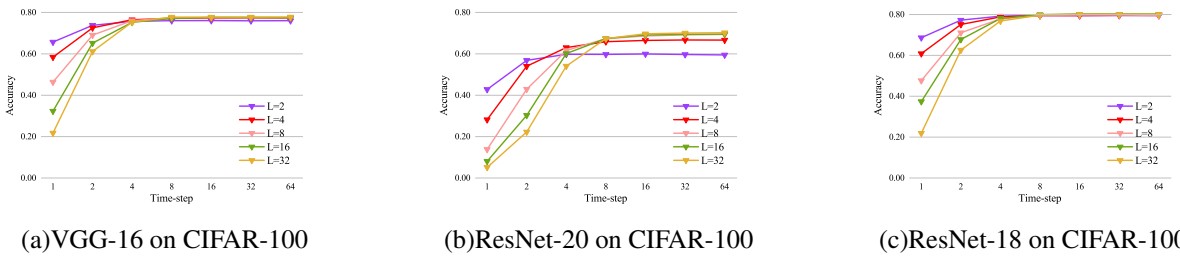

(a)VGG-16 on CIFAR-100    (b)ResNet-20 on CIFAR-100    (c)ResNet-18 on CIFAR-100

*Figure 6.* Influence of different quantization steps $L$ on CIFAR-100

*Table 7.* Influence of different quantization steps on the CIFAR-10 dataset

| Architacture | Quantization Steps | ANN | T=1 | T=2 | T=4 | T=8 | T=16 | T=32 | T=64 |
|---|---|---|---|---|---|---|---|---|---|
| VGG-16 | L=2 | 94.71% | 85.33% | 94.35% | 95.04% | 95.05% | 95.01% | 95.05% | 95.05% |
| | L=4 | 95.23% | 87.27% | 93.53% | 95.15% | 95.41% | 95.39% | 95.48% | 95.43% |
| | L=8 | 95.41% | 66.82% | 91.40% | 95.18% | 95.37% | 95.23% | 95.28% | 95.27% |
| | L=16 | 95.59% | 24.46% | 89.65% | 95.14% | 95.45% | 95.59% | 95.56% | 95.66% |
| | L=32 | 95.59% | 11.96% | 88.12% | 94.86% | 95.43% | 95.50% | 95.55% | 95.58% |
| ResNet-20 | L=2 | 88.56% | 78.77% | 87.54% | 89.83% | 90.07% | 89.95% | 90.08% | 90.11% |
| | L=4 | 91.76% | 67.31% | 84.52% | 90.55% | 92.10% | 92.40% | 92.43% | 92.46% |
| | L=8 | 92.89% | 51.79% | 82.78% | 90.96% | 92.58% | 92.95% | 93.03% | 93.03% |
| | L=16 | 93.37% | 33.88% | 66.05% | 88.74% | 92.95% | 93.32% | 93.34% | 93.41% |
| | L=32 | 93.53% | 20.18% | 54.94% | 86.15% | 92.24% | 93.29% | 93.42% | 93.41% |
| ResNet-18 | L=2 | 95.21% | 91.91% | 95.05% | 95.62% | 95.72% | 95.64% | 95.67% | 95.66% |
| | L=4 | 96.39% | 89.27% | 94.75% | 95.99% | 96.11% | 96.40% | 96.35% | 96.39% |
| | L=8 | 96.33% | 84.29% | 93.79% | 95.89% | 96.31% | 96.25% | 96.27% | 96.27% |
| | L=16 | 96.59% | 77.71% | 92.91% | 95.78% | 96.40% | 96.53% | 96.56% | 96.58% |
| | L=32 | 96.51% | 67.58% | 90.77% | 95.36% | 96.17% | 96.52% | 96.41% | 96.43% |

*Table 8.* Influence of different quantization steps on the CIFAR-100 dataset

| Architacture | Quantization Steps | ANN | T=1 | T=2 | T=4 | T=8 | T=16 | T=32 | T=64 |
|---|---|---|---|---|---|---|---|---|---|
| VGG-16 | L=2 | 74.63% | 65.70% | 73.78% | 75.70% | 76.01% | 76.06% | 75.95% | 76.01% |
| | L=4 | 76.46% | 58.39% | 72.58% | 76.50% | 77.14% | 77.29% | 77.39% | 77.28% |
| | L=8 | 77.32% | 46.37% | 69.07% | 76.03% | 77.41% | 77.44% | 77.46% | 77.45% |
| | L=16 | 77.67% | 32.32% | 65.19% | 75.24% | 77.09% | 77.66% | 77.58% | 77.50% |
| | L=32 | 77.83% | 21.73% | 61.12% | 75.32% | 77.74% | 77.75% | 77.77% | 77.82% |
| ResNet-20 | L=2 | 59.06% | 42.80% | 56.83% | 59.78% | 59.73% | 59.95% | 59.65% | 59.44% |
| | L=4 | 65.29% | 28.17% | 54.00% | 63.03% | 65.88% | 66.46% | 66.71% | 66.66% |
| | L=8 | 68.58% | 13.83% | 42.91% | 61.63% | 67.31% | 68.65% | 69.15% | 69.36% |
| | L=16 | 69.29% | 8.07% | 30.28% | 60.14% | 67.40% | 69.11% | 69.44% | 69.50% |
| | L=32 | 70.22% | 5.15% | 22.24% | 54.08% | 67.36% | 69.69% | 70.04% | 70.20% |
| ResNet-18 | L=2 | 77.75% | 68.65% | 77.23% | 79.21% | 79.49% | 79.44% | 79.49% | 79.52% |
| | L=4 | 79.09% | 60.82% | 75.11% | 78.81% | 79.48% | 79.45% | 79.55% | 79.50% |
| | L=8 | 79.44% | 47.62% | 71.09% | 77.84% | 79.59% | 79.73% | 79.73% | 79.49% |
| | L=16 | 80.33% | 37.39% | 67.77% | 78.02% | 79.95% | 80.25% | 80.26% | 80.24% |
| | L=32 | 80.04% | 21.89% | 62.60% | 76.80% | 79.69% | 80.10% | 80.04% | 80.03% |

