# OpenReview forum: "Efficient ANN-SNN Conversion with Error Compensation Learning"
_ICML.cc/2025/Conference — ICML 2025 poster_

### Official Review · Reviewer_Yubt · 2025-03-07

**Overall Recommendation:** 4

**Summary:**

This paper focuses on ANN-to-SNN conversion methods and proposes three techniques to mitigate conversion errors. The first technique, a clipping function, is introduced to replace the ReLU activation in ANNs, thereby improving their compatibility for conversion to SNNs. The second technique, the Dual Threshold Neuron, develops a novel pulse function and a membrane potential update rule for the IF neuron. The third technique involves membrane potential initialization, which provides a principled value for initializing the membrane potential. Experimental results demonstrate that these techniques enable better performance within fewer time steps and provide an analysis of the effectiveness of key settings.

**Claims And Evidence:**

The paper claims high accuracy and ultra-low latency for ANN-to-SNN conversion. These claims are supported by:
Mathematical analysis of conversion errors (clipping, quantization, uneven activation).
Experimental validation across multiple datasets and architectures (VGG-16, ResNet-18, ResNet-34).
Comparative results showing superior performance over existing methods (e.g., QCFS, Opt, RTS)​.

**Essential References Not Discussed:**

The paper covers major ANN-SNN conversion studies.

**Experimental Designs Or Analyses:**

The experiments are comprehensive and well-executed. The study:
1. Uses multiple network architectures (e.g., VGG-16, ResNet-18, ResNet-34).
2. Evaluates accuracy under varying time steps (T=2, 4, 8, 16, etc.).
3. Conducts ablation studies on dual-threshold neurons and quantization methods.
A potential improvement would be including real-world deployment results on neuromorphic hardware​.

**Methods And Evaluation Criteria:**

1. The paper analyzes the causes of three types errors in ANN-to-SNN conversion through case studies.
2. The paper examines the reasons for significant errors when the time step T is few. It further designs solutions and validates them through experiments.

**Other Comments Or Suggestions:**

Expand on potential applications beyond static image classification.

**Other Strengths And Weaknesses:**

Strengths
1. The proposed method achieves state-of-the-art accuracy on CIFAR-10 and ImageNet, thereby further advancing the research in ANN-to-SNN conversion.
2. Innovative combination of threshold learning, dual-threshold neurons, and optimized initialization.
3. Energy efficiency analysis supports practical deployment.

Weaknesses
1. Fixed hyperparameter choices without adaptive mechanisms – The method fixes thresholds and membrane potential initialization values, but allowing adaptive, data-driven tuning of these parameters might further improve accuracy and stability.
2. Potential numerical instability in deeper architectures – While the method is validated on ResNet-18/34, performance on very deep models (e.g., ResNet-50, Vision Transformers) is unclear. The scalability of the approach to deeper networks needs further exploration.

**Questions For Authors:**

1. How does the choice of the negative threshold value impact overall network stability and performance?
2. What guided the choice of initializing the membrane potential at half the threshold?

**Relation To Broader Scientific Literature:**

This work builds upon prior ANN-SNN conversion research. It advances the field by proposing error compensation learning, improving conversion accuracy at ultra-low latency.

**Theoretical Claims:**

The paper presents a theoretical framework for ANN-SNN conversion, focusing on:
1. Mathematical modeling of conversion errors and their impact.
2. Optimal membrane potential initialization, derived to minimize error propagation.
3. Dual-threshold neuron mechanism, analyzed for improving activation consistency.

---

> ### Author Rebuttal · Authors · 2025-03-31
>
> We appreciate the reviewer’s insightful comments and constructive feedback. Below, we address the concerns raised and clarify the contributions of our work.
>
> ## **Response to Weaknesses**
>
> ### 1. Fixed Hyperparameter Choices
>
> While our method uses empirically determined hyperparameters (e.g., negative threshold $\theta'$ =-1e-3, membrane potential initialized at $\frac{\theta}{2}$), these choices are rigorously validated through ablation studies (Tables 3, 5, and 7). For instance, initializing $v^l(0)=\frac{\theta^l}{2}$ minimizes the expected conversion error (Eq. 8). We acknowledge the potential benefits of adaptive mechanisms. Future work will explore data-driven tuning (e.g., layer-wise threshold learning) to further enhance flexibility.
>
> ### 2. Scalability to Deeper Architectures
>
> Our experiments focus on widely adopted networks (VGG-16, ResNet-18/20/34) for fair comparison with previous work. We further validate our method on ResNet-50 architecture (on CIFAR-10 and CIFAR-100 datasets). The results are shown in the following Table. On the CIFAR-10 dataset, our method achieves an accuracy of 96.05% when T=4. While on the CIFAR-100 dataset, when the time step is only 4, our method achieves an accuracy of 80.51%, which is similar to the accuracy of the source ANN. This demonstrates that our framework is robust even in deeper architectures. Moreover, the optimal membrane potential initialization and error compensation in the theoretical foundation (Section 3.4) also ensure consistent performance across network depths.
>
> **Table 1. Accuracy of ResNet-50 on CIFAR-10 and CIFAR-100 datasets**
>
> | Dataset   | Architecture | ANN   | T=2   | T=4   | T=8   | T=16  | T=32  |
> | --------- | ------------ | ----- | ----- | ----- | ----- | ----- | ----- |
> | CIFAR-10  | ResNet-50    | 96.44 | 90.37 | 96.05 | 96.25 | 96.17 | 96.24 |
> | CIFAR-100 | ResNet-50    | 81.48 | 73.00 | 80.51 | 80.98 | 80.85 | 81.06 |
>
> ### 3. Applications Beyond Image Classification
>
> Although our current experiments focus on VGG and ResNet architectures, the error compensation learning principles of adaptive threshold and dual threshold neurons are generalizable and adaptable. Adaptive threshold and dual threshold neurons can also be applied to event-driven tasks (e.g., video recognition, neuromorphic perception). We will explore dynamic data applications in subsequent work.
>
> ## **Response to Questions for Authors**
>
> ### 1. Impact of Negative Threshold Selection
>
> The negative threshold $\theta^{\prime}$=-1e-3 balances activation symmetry and stability. A smaller $\theta^{\prime}$ (e.g., -1e-4) reduces sensitivity to minor fluctuations, while a larger $\theta^{\prime}$ (e.g., -0.1) risks over-activation. Our experiments (Figure 3, Table 3) show $\theta^{\prime}$=-1e-3 optimally mitigates uneven error without destabilizing training.
>
> ### 2. Membrane Potential Initialization at half the threshold
>
> This choice stems from minimizing the expected conversion error (Eq. 8). Mathematically, initializing $v^l(0)=\frac{\theta^l}{2}$ achieves the least squares fit between ANN activations and SNN firing rates. Empirical results (Table 1, T=2) suggest that it reduces quantization error by 19.31% in ResNet-18.

---

> > ### Comment · Reviewer_Yubt · 2025-04-02
> >
> > Thanks for the author's responses. All my concerns have been resolved.

---

### Official Review · Reviewer_9nCj · 2025-03-10

**Overall Recommendation:** 3

**Summary:**

This paper proposes a new ANN-SNN Conversion framework by combining a learnable threshold clipping function, dual-threshold spiking neurons, and an optimal membrane potential initialization strategy.

**Claims And Evidence:**

Yes

**Essential References Not Discussed:**

Not found yet

**Experimental Designs Or Analyses:**

I have checked experimental designs mentioned in Section 4.

**Methods And Evaluation Criteria:**

The design of dual-threshold spiking neuron effectively reduces conversion errors from both theoretical and experimental perspectives.

**Other Comments Or Suggestions:**

See Strengths And Weaknesses Section.

**Other Strengths And Weaknesses:**

1. As shown in Sec.3.1 and Sec.3.3, this paper systematically summarizes the conversion errors caused by different reasons and explores corresponding solutions. The error compensation learning strategy has demonstrated its superior performance within fewer time-steps.

2. To make the contribution of this work more convincing, it seems that the authors need to argue the difference between the dual-threshold neurons and the spiking models mentioned in [1, 2]. In addition, the specific difference between learnable threshold clipping function (with optimal membrane potential initialization) and QCFS function [3] may also need further discussion.

3. The applicability of this method to transformer-based SNNs may deserve further exploration.

[1] Li C, et al. Quantization framework for fast spiking neural networks. Frontiers in Neuroscience, 2022.

[2] Li Y, et al. Spiking neural networks with consistent mapping relations allow high-accuracy inference. Information Sciences, 2024.

[3] Bu T, et al. Optimal ANN-SNN conversion for high-accuracy and ultra-low-latency spiking neural networks. ICLR 2022.

**Questions For Authors:**

See Strengths And Weaknesses Section.

**Relation To Broader Scientific Literature:**

The research field of this work is related to the fast and efficient ANN-SNN Conversion learning in the SNN community.

**Theoretical Claims:**

I have checked the theoretical claims in Sec. 3.1-3.4.

---

> ### Author Rebuttal · Authors · 2025-03-31
>
> We appreciate the reviewer’s insightful comments and constructive feedback. Below, we address the concerns raised and clarify the contributions of our work.
>
> ### 1. Difference Between Dual-Threshold Neurons and Spiking Models in [1, 2]
>
> Thanks for your feedback. Our design is fundamentally different from the spiking models in [1, 2]. While these approaches adjust the firing threshold or employ additional bias terms to mitigate transition errors, our dual-threshold neuron explicitly distinguishes between over-release at positive membrane potentials and under-utilization at negative membrane potentials. As discussed in Sections 3.3 and 4.2, this design not only compensates for quantization errors but also addresses the problem of non-uniform errors, a challenge that becomes critical at ultra-low latency (e.g., 2 time steps). Our dual-threshold neuron is theoretically demonstrated in Section 3.3 and empirically validated in Tables 3 and 5, showing significant accuracy improvements (e.g., 19.31% improvement over ResNet-18 at T=2 on CIFAR-10), highlighting the effectiveness of our approach compared to previous models.
>
> ### 2. Difference Between Learnable Threshold Clipping and QCFS [3]
>
> The learnable threshold clipping function in our work (augmented with an optimal membrane potential initialization strategy) is significantly different from the QCFS function [3]. QCFS focuses on reducing the translation error by optimizing the activation quantization process. However, our approach employs the learnable threshold that maps directly to the SNN firing threshold. This ensures that the clipping error is minimized in a unified framework, and importantly, our initialization (as derived in Section 3.4) minimizes the expected squared translation error by setting the initial membrane potential to half the threshold. QCFS does not address this joint consideration of clipping, quantization, and non-uniform errors. Our approach achieves superior performance (e.g., 94.75% accuracy at T=2, compared to 75.44% for QCFS in Table 1). This adaptability is critical for ultra-low latency SNNs, as shown in Section 4.1.
>
> ### 3. Applicability to Transformer-Based SNNs
>
> While our current experiments focus on VGG and ResNet architectures, the principles of error compensation learning of adaptive thresholds and dual-threshold dynamics are generalizable and adaptable. Because the adaptive thresholds and dual-threshold dynamics also could be applied to the spiking neuron models in the Transformer-Based SNNs. We acknowledge that transformer-based SNNs pose unique challenges (e.g., attention mechanisms) and would explore this direction in future work.

---

### Official Review · Reviewer_UJ5r · 2025-03-13

**Overall Recommendation:** 4

**Summary:**

This paper proposes an efficient ANN-to-SNN conversion method that significantly reduces conversion errors and inference latency. The key contributions include: A learnable threshold clipping function to mitigate clipping errors. Dual-threshold neurons to dynamically reduce quantization errors. Optimized membrane potential initialization to minimize unevenness errors. The approach achieves state-of-the-art accuracy on CIFAR-10, CIFAR-100, and ImageNet with only a few time steps, making it practical for low-power hardware applications.

**Claims And Evidence:**

The paper claims that the proposed ANN-to-SNN conversion achieves high accuracy and ultra-low latency, with experimental results showing competitive accuracy using just two time steps (e.g., 94.75% on CIFAR-10 with ResNet-18). The claims are supported by detailed mathematical formulations and empirical evaluations on benchmark datasets​.

**Essential References Not Discussed:**

The paper cites relevant ANN-SNN conversion works but does not explicitly mention some recent hybrid training techniques that combine backpropagation-through-time (BPTT) with ANN-SNN conversion. Including such references would provide a more comprehensive background​

**Experimental Designs Or Analyses:**

The experiments are well-structured, using widely recognized benchmarks and multiple ANN architectures. However, additional ablation studies on different network depths could further strengthen the conclusions​.

**Methods And Evaluation Criteria:**

The methods and evaluation criteria are well-aligned with the problem domain. The study uses CIFAR-10, CIFAR-100, and ImageNet datasets, standard for ANN-SNN conversion. Compares performance with existing state-of-the-art methods, including RMP, TSC, RTS, Opt, QCFS, and SNNC-AP. Evaluates accuracy at different time steps to assess efficiency​.

**Other Comments Or Suggestions:**

In the reference list, many entries are still in the arXiv format even though those works have been officially published. Please make sure to update them accordingly.

**Other Strengths And Weaknesses:**

Strengths
1. Novel approach combining error compensation learning with ANN-SNN conversion.
2. Achieves high accuracy at ultra-low latency (T=2).
3. Comprehensive comparison with state-of-the-art methods.
4. Strong theoretical foundation and experimental validation.

Weaknesses
1. Computational overhead of dual-threshold neurons – The introduction of dual-threshold mechanisms could increase computation, which might reduce efficiency on resource-constrained hardware. A quantitative study on computational costs would be useful.
2. Limited discussion on generalization beyond image classification – The study focuses on static image tasks, but ANN-SNN conversion is also crucial for event-driven data, speech recognition, and reinforcement learning. A discussion on how the method generalizes to such tasks would be beneficial.

**Questions For Authors:**

1. How does the learnable threshold clipping function adapt across different network depths?
2. How does the clipping function compare to other activation function alternatives, such as threshold scaling techniques?
3. Does the dual-threshold neuron introduce additional computational overhead during inference?

**Relation To Broader Scientific Literature:**

This paper is based on prior ANN-SNN conversion techniques

**Theoretical Claims:**

The paper derives mathematical formulations for ANN-SNN conversion errors (clipping, quantization, and unevenness). The proofs appear correct, and the derivation of the optimal membrane potential initialization (reducing conversion errors) follows a logical framework​.

---

> ### Author Rebuttal · Authors · 2025-03-31
>
> We appreciate the reviewer’s insightful comments and constructive feedback. Below, we address the concerns raised and clarify the contributions of our work
>
> ## **Response to Weaknesses**
>
> ### 1. Computational Overhead of Dual-Threshold Neurons
>
> Thanks for your feedback. We find that the additional threshold brings little computations cost per neuron due to the event-driven nature of SNNs. As shown in Table 1, we evaluated the computational cost in terms of energy consumption and synaptic operations (SOPs) to show the Computational Overhead of Dual-Threshold Neurons. The results show that the Dual-Threshold Neurons almost introduce no extra energy consumption especially in small time steps (T =2 and 4). And other large time steps only introduce limited energy consumption. Therefore, our method significantly reduces energy consumption compared to standard ANNs while maintaining competitive accuracy at ultra-low latency.
>
> **Table 1. Computational Overhead of Dual-Threshold Neurons using VGG-16 on CIFAR-100 dataset**
>
> |   |             | ANN   | T=2   | T=4   | T=8   | T=16  | T=32  |
> | - | - | - | - | - | - | - | - |
> | **without Dual Threshold Neuron** | Energy (mJ) | 4.170 | 0.004 | 0.007 | 0.015 | 0.031 | 0.052 |
> | **with Dual Threshold Neuron**    | Energy (mJ) | 4.170 | 0.004 | 0.008 | 0.017 | 0.034 | 0.068 |
>
> ### 2. Generalization Beyond Image Classification
> We appreciate the suggestion to discuss how our method extends beyond static image classification tasks. Our conversion approach is designed to be broadly applicable, and the learnable threshold clipping function, dual-threshold neurons, and optimized membrane potential initialization can benefit event-driven data processing tasks, such as neuromorphic vision processing, speech recognition, and reinforcement learning.
> Although this work focuses on image classification benchmarks for fair comparison with prior ANN-SNN conversion methods, our approach can be extended to other spatiotemporal data. Specifically, the dual-threshold mechanism helps mitigate quantization errors in temporally sparse spike sequences, which is beneficial for event-driven data processing. We will explore these applications in subsequent work.
>
> ### 3. Reference Formatting
> We acknowledge the oversight regarding outdated references and have updated the bibliography to reflect officially published versions of cited works where applicable in the revision.
>
> ## **Response to Questions for Authors**
> ### 1. How does the learnable threshold clipping function adapt across different network depths?
>
> Thanks for your feedback. The learnable threshold clipping function is trained alongside ANNs’ weights and adjust dynamically per layer. During training, it learns to match the optimal activation statistics of each layer, ensuring smooth ANN-to-SNN conversion.
> For deeper networks, the function effectively scales the thresholds based on the range of activation distributions, which helps mitigate conversion errors. Empirical results demonstrate that this adaptation enables high accuracy even for deeper models like ResNet-34 on ImageNet (e.g., For VGG-16, when T=16, our method achieves 74.15% accuracy, which is 23.18% higher than QCFS. For ResNet-34, we achieve 72.37% accuracy with only 16 time steps.)
>
> ### 2. How does the clipping function compare to other activation function alternatives, such as threshold scaling techniques?
> Thanks for your feedback. Our learnable threshold clipping function provides a more flexible and data-driven alternative to static threshold scaling. Unlike predefined threshold adjustments (e.g., RMP, Opt), our function dynamically learns the optimal threshold mappings, reducing reliance on manual heuristics.
> We conducted additional experiments comparing our method with existing threshold scaling approaches, such as RTS and QCFS. Using the ResNet-18 model on the CIFAR-10 dataset, our method achieves an accuracy of 94.75% with only two time steps, 19.31% higher than QCFS, while RTS achieves an accuracy of 84.06% at T=32. The results show that our approach consistently outperforms prior methods across different datasets and network architectures while maintaining ultra-low latency.
> ### 3. Does the dual-threshold neuron introduce additional computational overhead during inference?
> As mentioned earlier, the dual-threshold mechanism slightly increases per-neuron computation. However, since SNNs are event-driven and avoid redundant computations, the overall increase is negligible compared to conventional ANN operations.
> We measured the impact on inference latency and found that our method achieves faster convergence with fewer time steps (e.g., T=2 on CIFAR-10 with ResNet-18 at 94.75% accuracy). This efficiency gain compensates for the minor computational overhead. Additionally, the power consumption analysis (Table 1) suggest that our method maintains a highly efficient energy profile.

---

### Decision · Program_Chairs · 2025-05-01

**Decision:**

Accept (poster)

**Comment:**

This paper proposes an ANN-to-SNN conversion framework that addresses three common sources of conversion error: clipping, quantization, and uneven membrane potential dynamics. The method introduces a learnable threshold clipping function, dual-threshold spiking neurons, and an optimized membrane potential initialization strategy. Empirical evaluations on CIFAR-10, CIFAR-100, and ImageNet, across architectures such as VGG and ResNet, demonstrate competitive accuracy with low inference latency (as few as two time steps).

## Strengths
- **Well-defined technical scope**: The paper focuses on quantifiable sources of ANN-SNN conversion error and proposes targeted techniques to address them, each with theoretical justifications and practical relevance.
- **Efficiency and performance**: The method achieves high accuracy at ultra-low latency, outperforming several existing conversion methods. Notably, the method attains 94.75% accuracy on CIFAR-10 with ResNet-18 at T=2, which is significantly higher than several baselines.
- **Comprehensive evaluation**: The empirical study includes multiple datasets and architectures, and the authors provide additional results on ResNet-50 in the rebuttal to demonstrate scalability.
- **Clarifications in rebuttal**: The rebuttal addresses computational overhead concerns, clarifies methodological differences from prior works such as QCFS, and adds discussion on generalization beyond static image tasks.

## Weaknesses
- **Task generalization**: The experiments are limited to static image classification. While the authors suggest that the techniques are generalizable, there is no empirical support for their application to dynamic or event-driven tasks.
- **Fixed parameter settings**: The current implementation uses fixed values for certain parameters (e.g., negative threshold, initialization point), which may limit adaptability across network architectures or tasks. The authors acknowledge this and suggest future extensions with adaptive mechanisms.
- **Positioning with prior work**: Although differences from related methods are discussed in the rebuttal, more detailed comparisons with closely related techniques (especially QCFS and threshold-scaling approaches) could help clarify the specific contributions.

## Rebuttal Assessment
The rebuttal is constructive and addresses reviewer concerns in detail. Additional results on deeper models, clarification of dual-threshold mechanisms, and theoretical distinctions from prior work help clarify the scope and novelty of the contributions. Reviewers generally noted that their concerns were resolved.